# Engineering a transposon-associated TnpB-ωRNA system for efficient gene editing and phenotypic correction of a tyrosinaemia mouse model

Zhifang Li [1,10], Ruochen Guo[1,2,10], Xiaozhi Sun[1,3,10], Guoling Li[4,10], Zhuang Shao[1], Xiaona Huo[1,5], Rongrong Yang[1,5], Xinyu Liu[2], Xi Cao[1,6], Hainan Zhang[4], Weihong Zhang[4], Xiaoyin Zhang[1,5], Shuangyu Ma[7], Meiling Zhang[8], Yuanhua Liu[2], Yinan Yao[2], Jinqi Shi[1], Hui Yang [2,4,5], Chunyi Hu [9] ✉, Yingsi Zhou [4] ✉ & Chunlong Xu [1,3,5] ✉

Transposon-associated ribonucleoprotein TnpB is known to be the ancestry endonuclease of diverse Cas12 effector proteins from type-V CRISPR system. Given its small size (408 aa), it is of interest to examine whether engineered TnpB could be used for efficient mammalian genome editing. Here, we showed that the gene editing activity of native TnpB from *Deinococcus radiodurans* (ISDra2 TnpB) in mouse embryos was already higher than previously identified small-sized Cas12f1. Further stepwise engineering of noncoding RNA (ωRNA or reRNA) component of TnpB significantly elevated the nuclease activity of TnpB. Notably, an optimized TnpB-ωRNA system could be efficiently delivered in vivo with single adeno-associated virus (AAV) and corrected the disease phenotype in a tyrosinaemia mouse model. Thus, the engineered miniature TnpB system represents a new addition to the current genome editing tool-box, with the unique feature of the smallest effector size that facilitate efficient AAV delivery for editing of cells and tissues.

The TnpB proteins represent a family of transposon-associated RNA-guided endonucleases. Recent biochemical studies[1,2] revealed that TnpB proteins are ancestry predecessors of Cas12 effector proteins in the type-V CRISPR system, and a 247-nucleotides (nt) noncoding RNA (termed ωRNA or reRNA) derived from the right end of transposon element is the required component for ISDra2 TnpB to recognize and cleave target DNA. The size of TnpB proteins, with ~400 amino acid (aa) residues, is much smaller than their evolutionary progeny Cas12 proteins (mostly ~1000 aa). Furthermore, in vitro studies[1,2] demonstrated that TnpB exhibited double-strand DNA cleavage activity

[1]Lingang Laboratory, Shanghai, China. [2]Institute of Neuroscience, State Key Laboratory of Neuroscience, Key Laboratory of Primate Neurobiology, CAS Center for Excellence in Brain Science and Intelligence Technology, Shanghai Institutes for Biological Sciences, Chinese Academy of Sciences, Shanghai, China. [3]School of Life Sciences and Technology, ShanghaiTech University, Shanghai, China. [4]HuidaGene Therapeutics Inc, Shanghai, China. [5]Shanghai Center for Brain Science and Brain-Inspired Technology, Shanghai, China. [6]College of Animal Science and Technology, Northwest A&F University, Yangling, China. [7]Department of Histoembryology, Genetics and Developmental Biology, Shanghai Key Laboratory of Reproductive Medicine, Shanghai JiaoTong University School of Medicine, Shanghai, China. [8]Center for Reproductive Medicine, International Peace Maternity and Child Health Hospital, Innovative Research Team of High-level Local Universities in Shanghai, School of Medicine, Shanghai Jiao Tong University, Shanghai, China. [9]Department of Biological Sciences, National University of Singapore, Singapore, Singapore. [10]These authors contributed equally: Zhifang Li, Ruochen Guo, Xiaozhi Sun, Guoling Li. ✉e-mail: hu_dbs@nus.edu.sg; yingsizhou@huidagene.com; xucl@lglab.ac.cn

guided by ωRNA. Therefore, there is potential for the use of this TnpB system in genome editing and therapeutic applications.

Gene editing using Cas9 or Cas12 systems has been widely used in animal models and recently applied in clinical trials. At present, AAV is the most commonly used delivery system and shown to be safe in gene therapy[3]. However, the maximal cargo size of AAV was limited to be 4.7 kilobase (kb) pairs, hindering efficient in vivo delivery of the large Cas9 or Cas12 protein via single AAV injection. This size problem is exacerbated in the use of base and prime editors comprising Cas9 (or Cas12) and fusion enzymes. Recent identification of compact CRISPR effector proteins Cas12f1 (~500 aa)[4] and Cas13 (~700 aa)[5,6] represent potential solutions. However, the gene editing efficiency of Cas12f1 was relatively low[7–11], whereas Cas13 exhibited collateral RNA cleavage activity with uncertain safety profile[12,13].

In this study, we demonstrated that genome editing activity of TnpB was markedly higher than that of Cas12f1 in cultured cells and mouse embryos. To further optimize the TnpB system, we engineered TnpB-associated ωRNA in a stepwise manner to identify the optimal ωRNA variant with the shortest sequence length and elevated gene editing activity. Importantly, we showed that the optimized TnpB-ωRNA system could be effectively delivered in vivo via a single AAV injection in tyrosinaemia model mice, leading to the correction of the disease phenotype. Thus, we have shown the applicability of the engineered hypercompact TnpB for genome editing in vivo.

## Results

### TnpB exhibited gene editing activity higher than Cas12f1

Previous studies have shown the endonuclease activity of several Cas12f1 orthologs from type-V-U CRISPR family that have small sizes. As the ancestry enzyme of Cas12 proteins, TnpB (~400 aa) represents the smallest programmable nuclease among common single effector Cas proteins, including SpCas9, LbCas12a, Un1Cas12f1, and IscB (Fig. 1a). However, the mammalian genome editing potential of TnpB remained to be fully characterized. Thus, we selected several genomic loci to evaluate the editing activity of ISDra2 TnpB (hereafter as TnpB) from *Deinococcus radiodurans* in mouse embryos. First, we in vitro transcribed ωRNA that targets the mouse *Tyr* gene (Fig. 1b), and injected ωRNA together with TnpB mRNA into mouse embryos. The injected embryos were then transferred into surrogate female mice to generate gene-modified offspring. Since *Tyr* gene encodes the black coat color of C57BL/6 mice, we estimated the efficiency of TnpB-induced gene disruption by directly examining the coat color change in TnpB-injected mice. We found that TnpB treatment completely converted black coat color into albino white in all newborn mice (Fig. 1c). In contrast, similar embryo injection of Un1Cas12f1 together with sgRNA targeting the *Tyr* gene did not change the black coat color in the newborn mice (Fig. 1c), suggesting a much lower *Tyr* gene disruption efficiency of Un1Cas12f1 than that of TnpB. Further deep-sequencing for *Tyr* gene showed that 20% and 90% of indel mutations were induced by Un1Cas12f1 and TnpB, respectively (Fig. 1b and Supplementary Fig. S1). Although Cas12f1 and TnpB have different requirements for target adjacent motif (TAM, also known as PAM) that recognizes the target sequence, we have chosen the targeted sequence in *Tyr* gene to have 17-bp overlap (among 20 bp) for both enzymes (Fig. 1b). Thus, the higher editing efficiency of TnpB as compared to Cas12f1 was largely due to its intrinsic activity.

To further evaluate the gene editing activity of TnpB, we chose six additional loci in the mouse *Dmd* gene (Fig. 1d) for targeting in mouse embryos, by injecting ωRNA targeting these loci with TnpB mRNA. As shown by deep-sequencing results, TnpB exhibited an average of 90% editing efficiency for all six targeted loci in the *Dmd* gene (Fig. 1d and Supplementary Figs. S2 and S3). Furthermore, the gene editing outcome was verified by immunostaining of dystrophin protein encoded by *Dmd* gene that is specifically expressed in muscle tissues. In contrast to wild-type mice, TnpB-treated mice showed undetectable dystrophin

expression in heart, Diaphragm (DI) and Tibialis anterior (TA) muscles (Fig. 1e and Supplementary Fig. S4), suggesting the complete disruption of *Dmd* gene by TnpB and ωRNA injection. Finally, these immunostaining results were confirmed by western blotting of dystrophin protein of various muscle tissues (Fig. 1f). Consequently, rotarod and grip strength assessment of TnpB-treated DMD mice found functional dysfunction of muscle (Supplementary Fig. S5). Thus, our finding indicated more robust gene editing activity of TnpB than that of Un1Cas12f1 in mammalian tissues.

### Engineered TnpB-associated ωRNA with elevated editing efficiency

Cognate ωRNA scaffold associated with TnpB is 231 nt, much longer than sgRNA scaffold for most single effector Cas proteins. Previous findings reported that the sgRNA engineering could improve the performance of gene editing enzymes[14]. We thereby hypothesized that ωRNA truncation and optimization might be helpful for enhancing TnpB activity in mammalian cells. To this end, we predicted the secondary structure of ωRNA and formulated a stepwise strategy to truncate ωRNA (Fig. 2a). Based on the stem loops in predicted structure, we divided ωRNA into six segments, named as S1 to S6 for the truncation experiment (Fig. 2b). To facilitate screen of ωRNA variants, we designed a gene editing reporter with TnpB target DNA placed within a split and frameshift GFP gene which could only be repaired via the single-strand annealing (SSA) pathway[15] after disruption of TnpB target sequence to express GFP (Fig. 2a). We tested the reporter with cognate ωRNA to prove the conditional activation of GFP after treatment of TnpB guided by ωRNA targeting frameshift mutation in GFP gene (Fig. 2a). At first, we deleted S1 to S6 one by one and run the reporter assay. It showed that only deletion of S4 and S6 ablated the activity of TnpB (Fig. 2c), suggesting the dispensable role of S1, S2, S3, and S5 for normal ωRNA function. Furthermore, sequence deletion of S1 slightly increase TnpB activity (Fig. 2c).

To interrogate combined deletion effect of S1 to S6, we added S2 to S5 deletion in the S1 deletion variant of ωRNA to conduct reporter assay. It found that simultaneous deletion of S1, S2, and S3 in ωRNA-v1 not only supported the normal function of TnpB but also significantly increased the gene editing efficiency (Fig. 2d). These results implied that the ωRNA sequence from S4 to S6 dictated the enzymatic activity of TnpB. Secondary structure of ωRNA after combined truncation of S1, S2, and S3 showed typical stem-loop conformations with three distinguishable and consecutive stem-loop (SL) domains, termed as SL1, SL2, and SL3 (Fig. 2e). To further determine the effect of these three SL domains on TnpB activity, we iteratively remove SL1, SL2, and SL3 for reporter test. In addition, we also generated two other ωRNA variants with partial deletion of SL2 subdomain or substitution of G:U with G:C pairs (Fig. 2e). We found that SL1, SL2 and SL3 are necessary for the normal function of TnpB since deletion variants lack of any single SL fully blocked the reporter activation (Fig. 2f). However, partial replacement of SL3 subdomain with 5′-GAAA-3′ loop sequence actually enhanced the TnpB activity (Fig. 2f). G:C substitution for G:U pair exhibited no additive effect on the performance of TnpB (Fig. 2f). Based on these results, we finally identified an optimal ωRNA variant ωRNA-v2 or ωRNA* that improved TnpB performance. The predicted secondary structure of ωRNA* presented with three compact stem-loop domains in contrast to the loose organization of cognate ωRNA structure (Fig. 2g).

### Characterization of endogenous gene editing and off-target activity for TnpB-ωRNA system

To verify the reporter assay results for ωRNA*, we selected 14 endogenous genomic loci for further evaluation of gene editing performance in HEK293T (Fig. 3a). Among 14 human loci tested, 10 individual target sites showed significant increase of TnpB gene editing efficiency with ωRNA* compared to original ωRNA (Fig. 3b). Moreover, we also

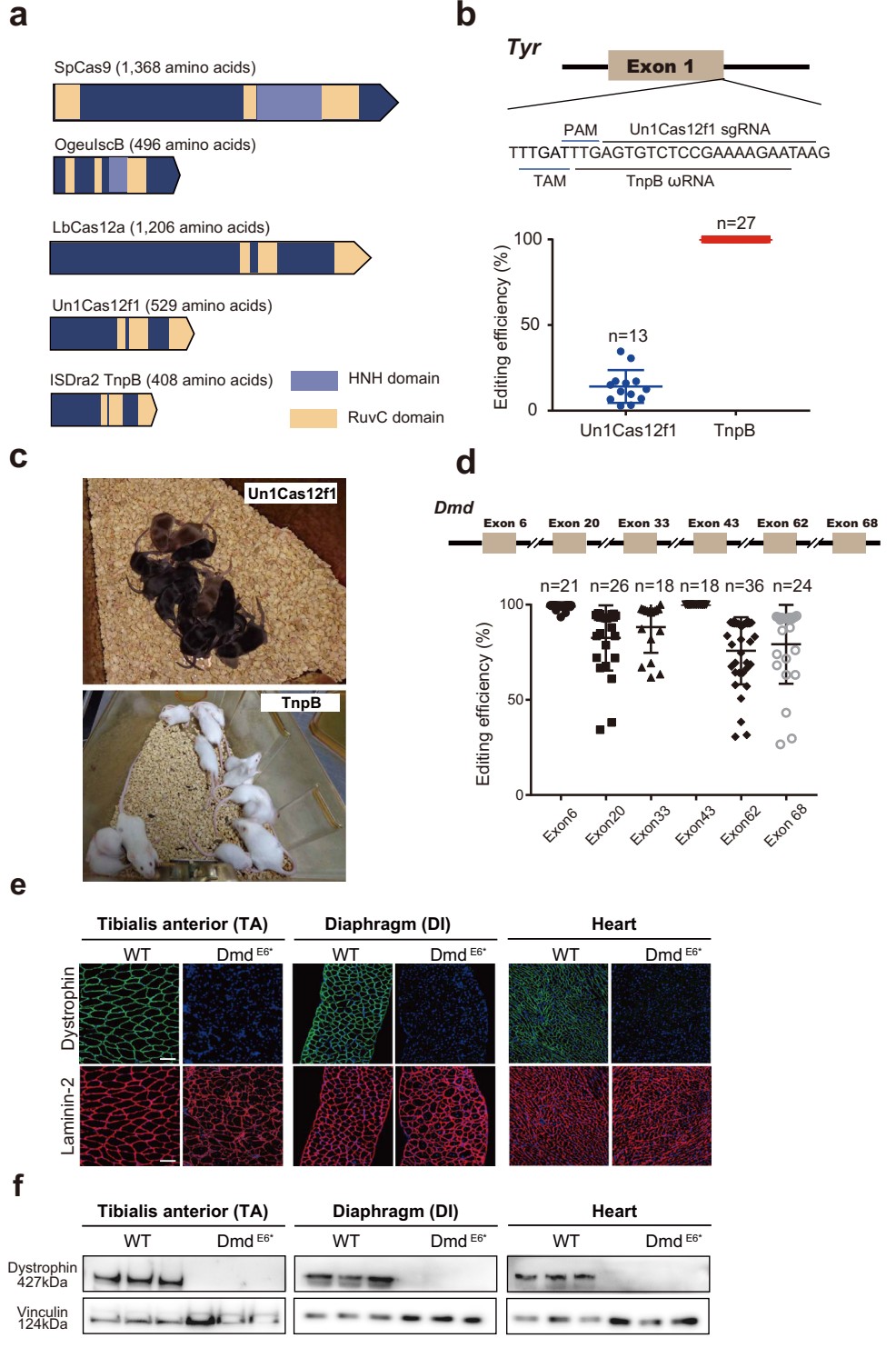

**Fig. 1 | Mouse embryo injection of TnpB-ωRNA induced efficient gene editing.**
**a** Characteristics of SpCas9, IscB, LbCas12a, Un1Cas12f1 and ISDra2 TnpB nucleases.
**b** Comparison of editing efficiency between TnpB and Cas12f1 on *Tyr* gene in mice.
**c** Coat color phenotype of *Tyr* gene-edited mice by Un1Cas12f1 and TnpB. **d** TnpB-mediated gene editing efficiency for *Dmd* gene. **e** Dystrophin and laminin-2 immunostaining for TA, DI, and heart muscle tissues in wild-type and *Dmd*-edited mice by TnpB. **f** Western blotting of dystrophin and vinculin protein for three muscle tissues in wild-type and *Dmd*-edited mice by TnpB. Data are represented as means ± SEM. A dot represents a biological replicate ($n = 3$ or more). Unpaired two-tailed Student's *t* tests. Significant differences between conditions are indicated by asterisk (*$P < 0.05$, **$P < 0.01$, ***$P < 0.001$, ****$P < 0.0001$, NS non-significant). Scale bars, 100 µm. Source data are provided as a Source Data file.

compared the gene editing activity of TnpB-ωRNA and -ωRNA* with that of SaCas9 and SpCas9 in both mouse N2a and human HEK293T cells. Our results showed that TnpB-ωRNA* exhibited similarly high efficiency with SaCas9 but only slightly lower activity than SpCas9 (Fig. 3c and Supplementary Figs. S6–9). To investigate broad improvement effect of ωRNA* in mammalian cells, we further performed the gene editing in mouse N2a cells targeting four disease-relevant genes, including *Klkb1*, *Tyr*, *Hpd* and *Pcsk9*. It found that all genomic sites exhibited significantly increased gene editing efficiency for ωRNA* compared to the original ωRNA (Supplementary Fig. 10).

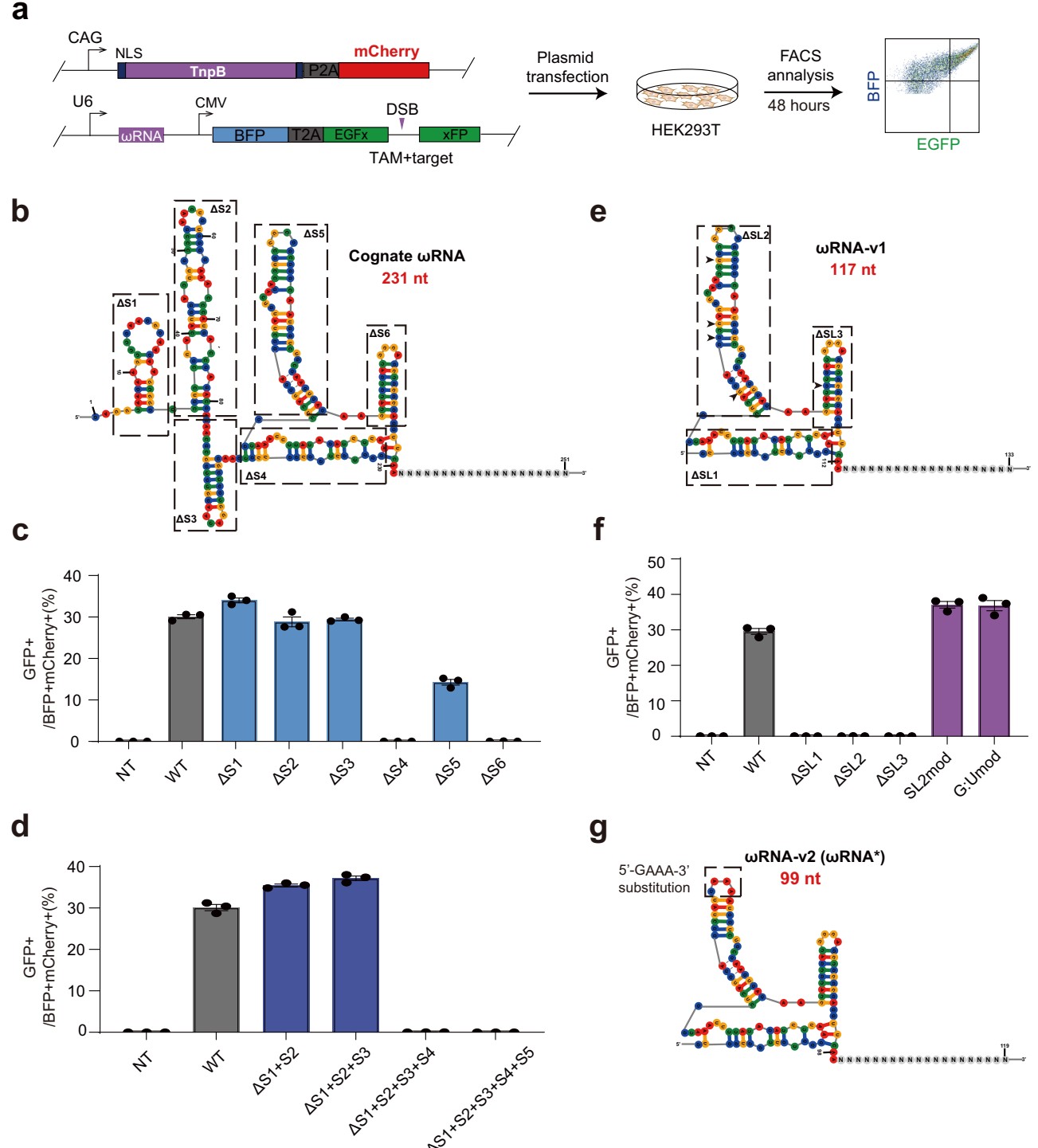

**Fig. 2 | Stepwise engineering of TnpB-associated ωRNA improved gene editing efficiency. a** Reporter assay schematics of detecting cleavage activity of TnpB-ωRNA. **b** Predicted secondary structure of cognate ωRNA (231 nt). Cognate ωRNA was divided into 6 segments, named from S1 to S6. **c** Reporter assay results using engineered ωRNA by one-by-one truncation of S1 to S6. **d** Reporter assay results with engineered ωRNA by different combined truncations of S1 to S5. **e** Predicted secondary structure of a ωRNA-v1 variant with simultaneous truncation of S1, S2, and S3. **f** Reporter assay results for ωRNA variants with different SL deletion and modifications. **g** Predicted secondary structure of final optimized ωRNA-v2 (ωRNA*) variant. Data are represented as means ± SEM. A dot represents a biological replicate (*n* = 3). Source data are provided as a Source Data file.

Quantitative analysis revealed twofold increase of gene editing efficiency in N2a for ωRNA* versus wild-type ωRNA (Supplementary Fig. 10). In particular, the ωRNA* even supported TnpB editing of some loci that are edited with very low efficiency using the cognate ωRNA (Supplementary Fig. 10). Furthermore, we packaged TnpB-ωRNA and -ωRNA* with AAV to evaluate their gene editing efficiency in mice, which also confirmed the significantly improved performance with TnpB-ωRNA* in vivo (Supplementary Fig. 11). Therefore, we demonstrated the enhanced TnpB activity in both mammalian cells and mice via the identification of ωRNA* after stepwise engineering.

To examine the off-target effect of TnpB, we carried out prediction of potential off-target genomic loci with Cas-OFFinder[16] for off-target analysis when designing ωRNA against a target site in *Hpd* gene (Fig. 3d). For the top ten predicted off-target sites, no gene editing events was detected for TnpB-ωRNA targeting *Hpd* (Fig. 3d). Furthermore, we also performed genome-wide off-target analysis by PEM-seq[17] to identify potential translocation between on-target and off-target loci. Our PEM-seq results showed that there is no induction of translocation events related to gene editing of *Hpd* gene by the engineered TnpB-ωRNA treatment (Fig. 3e).

## Correction of fatal liver disease with in vivo delivery of TnpB-ωRNA via single AAV

Given the hypercompact size of TnpB, it would greatly facilitate in vivo delivery via single AAV for gene editing therapy. To demonstrate the potential of TnpB in disease intervention, we chose the *Hpd* as therapeutic target for gene editing therapy of type I hereditary tyrosinaemia (HT1) in Fah[−/−] mouse model. Adult Fah[−/−] was administrated with AAV-TnpB (TnpB only without ωRNA), AAV-TnpB-ωRNA, or AAV-TnpB-ωRNA* (Fig. 4a) and kept without nitisinone (NTBC) drug, an inhibitor of 4-hydroxyphenylpyruvate dioxygenase (HPD) to examine

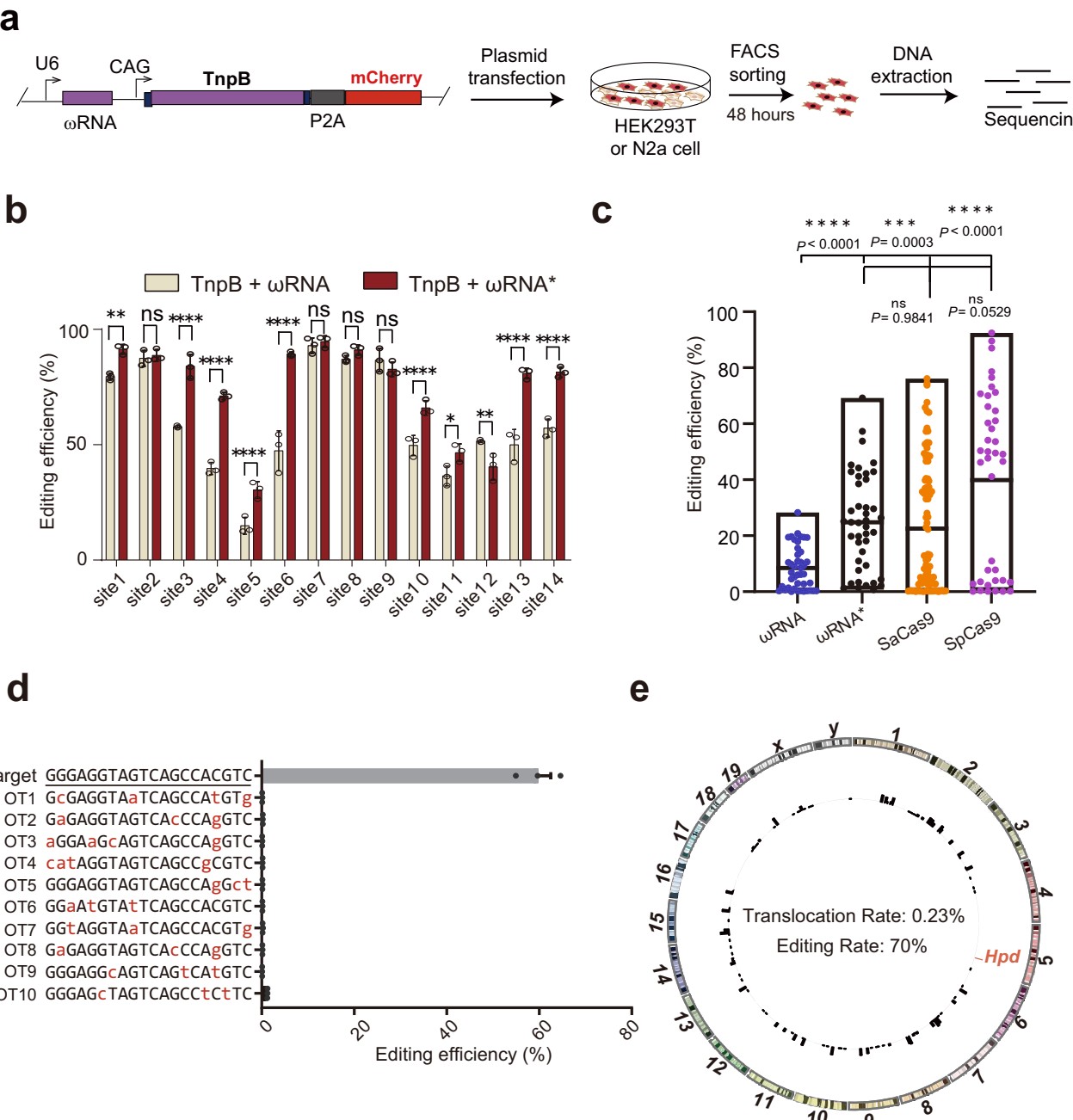

**Fig. 3 | Characterization of endogenous gene editing activity and off-target effect with optimized TnpB-ωRNA system. a** The experimental workflow for detecting editing efficiency of original and optimized TnpB-ωRNA in HEK293T cells. **b** Gene efficiency comparison results for 14 human endogenous gene loci targeted by wild-type and optimized TnpB-ωRNA in HEK293T. **c** Gene editing activity comparison among TnpB-ωRNA, TnpB-ωRNA*, SaCas9 and SpCas9. All data points with median at center line and 25th and 75th quartile lines. Detailed statistical results in Source Data file. **d** Off-target analysis for top predicted off-target genomic loci via Cas-OFFinder. **e** Genome-wide off-target analysis with PEM-seq for the engineered TnpB-ωRNA. Data are represented as means ± SEM. A dot represents a biological replicate (*n* = 3 or more). Unpaired two-tailed Student's *t* tests. Significant differences between conditions are indicated by asterisk (*P < 0.05, **P < 0.01, ***P < 0.001, ****P < 0.0001, NS non-significant.). Source data are provided as a Source Data file.

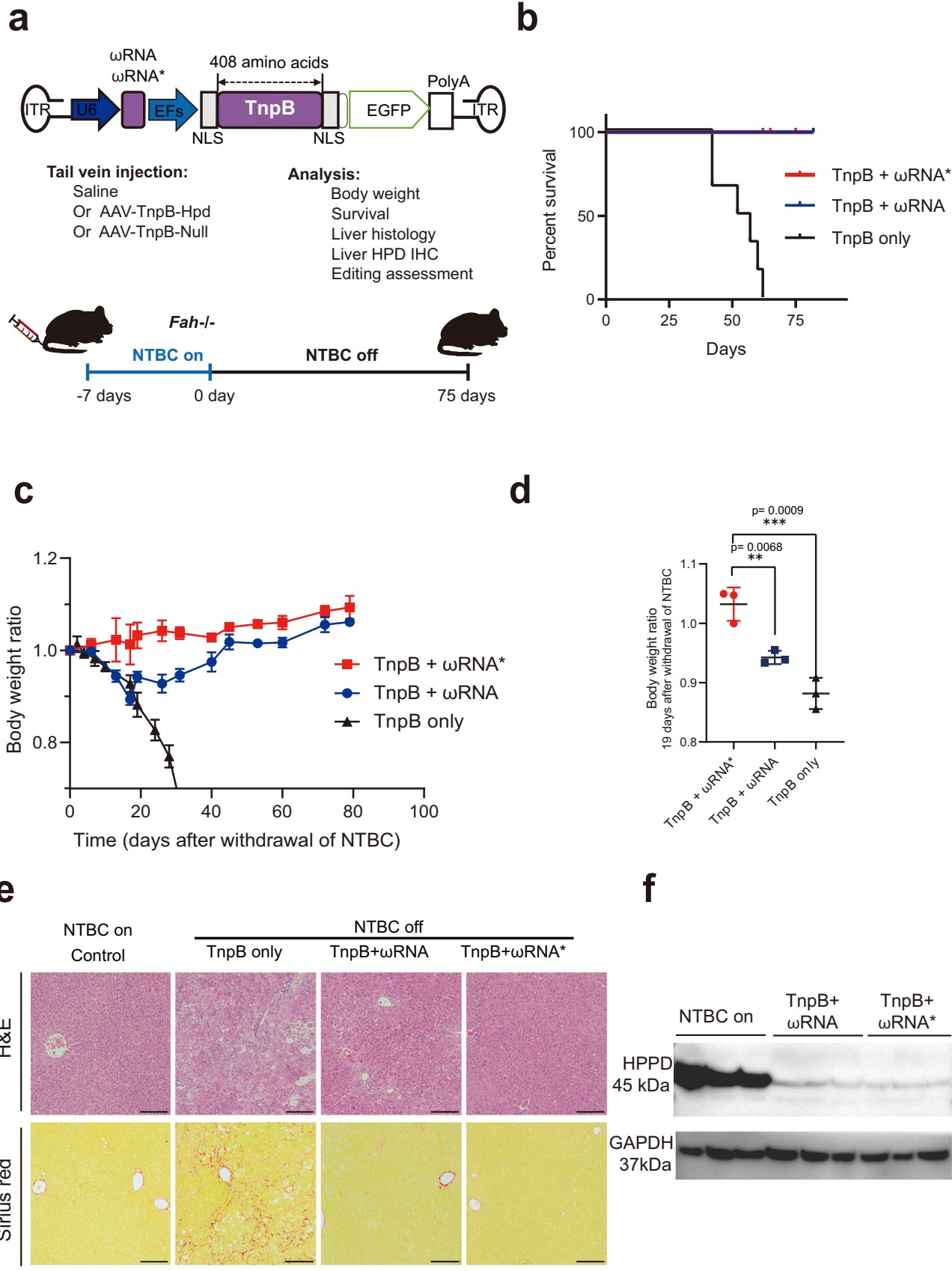

**Fig. 4 | Correction of fatal liver disease with in vivo delivery of TnpB-ωRNA via single AAV. a** Diagram of AAV-TnpB-ωRNA/ωRNA* vector and gene therapy schematics in Fah⁻/⁻ mouse model of type I hereditary tyrosinaemia. **b** Survival curve for disease mice treated with AAV-TnpB-ωRNA/ωRNA* or AAV-TnpB without ωRNA (TnpB only group). **c** Body weight change during the observation period for disease mice in different treatment groups. **d** Body weight ratio for TnpB-ωRNA or TnpB only versus TnpB-ωRNA*-treated mice in 19-day after NTBC withdrawal.

**e** Histology analysis with H&E and Sirius red staining for mouse liver from different treatment groups. **f** Western blot for HPD protein from untreated and TnpB-ωRNA-treated HT1 mice. Data are represented as means ± SEM. A dot represents a biological replicate (*n* = 3 or more). Unpaired two-tailed Student's *t* tests. Significant differences between conditions are indicated by an asterisk (*$P < 0.05$, **$P < 0.01$, ***$P < 0.001$, ****$P < 0.0001$, NS non-significant.). Scale bars, 200 μm. Source data are provided as a Source Data file.

the therapeutic effect after gene editing. We observed that AAV-TnpB-ωRNA- and -ωRNA*-treated Fah⁻/⁻ mice was still alive after 75 days without NTBC, but all mice in TnpB only group died at about 65 days (Fig. 4b). Furthermore, Fah⁻/⁻ mice subjected to AAV-TnpB-ωRNA treatment gained body weight after experiencing a short period of weight loss (Fig. 4c, d). Notably, mice with AAV-TnpB-ωRNA* treatment did not experience weight loss after NTBC withdrawal (Fig. 4c, d), indicating better therapeutic efficacy than that of wild-type TnpB-ωRNA. In addition, we found that Un1Cas12f1-treatment could also rescue HT1 mice but with significantly decreased therapeutic efficacy compared to AAV-TnpB-ωRNA and -ωRNA* in terms of body weight loss (Supplementary Fig. 12). Contrarily, mice treated without ωRNA exhibited rapid weight loss until death (Fig. 4c and Supplementary Fig. 12). These results suggested the importance for improving gene-editing activity with the engineered ωRNA to increase the therapeutic effect for disease correction. To examine the histological correction by gene editing treatment, we performed H&E and Sirius red staining to find dramatically reduced fibrotic pathology in TnpB-ωRNA/ωRNA* treated mice, whereas massive liver fibrosis in mice treated without ωRNA (Fig. 4e).

Furthermore, we also analyzed the HPD expression in treated versus untreated mice. It showed the remarkable decrease of HPD protein (Fig. 4f) and large HPD-negative liver region in AAV-TnpB-ωRNA-treated mice (Supplementary Fig. 13a, b). To investigate the in vivo gene editing outcomes, we collected liver tissue from mice for deep-sequencing analysis. We found 20% and 60% of indel rate for AAV-TnpB-ωRNA- and AAV-TnpB-ωRNA*-treated mice in 1 month, respectively (Supplementary Fig. 13c, d). Consistently, liver metabolic functions were significantly ameliorated after AAV-TnpB-ωRNA treatment as indicated by the blood biochemical profiling results of alanine aminotransferase (ALT), aspartate aminotransferase (AST), total bilirubin and tyrosine (Supplementary Fig. 14a–d). Moreover, we analyzed the succinylacetone level in TnpB-ωRNA-treated mice and confirmed that TnpB-ωRNA treatment largely reduced succinylacetone in both plasma and urine of treated mice (Supplementary Fig. 14e, f), consistent with phenotypic correction of HT1 mice after TnpB-ωRNA administration. Therefore, our results showed the proof-of-concept for applying the engineered TnpB-ωRNA system in disease control via single AAV delivery in vivo.

## Discussion

Diverse CRISPR-Cas systems evolved from immune battle between microbe and mobile genetic elements (MGE), providing us abundant resources for the identification of gene editing enzymes[18]. In the past years, various single effector Cas proteins including Cas9[19], Cas12[20], and Cas13[21] were found to deploy DNA or RNA editing activity in different organisms for both research and therapeutic purpose[22]. Recently, TnpB-like proteins, including IscB and TnpB associated with microbe transposon element, were identified to be active ancestry endonuclease for Cas9 and Cas12[1,2]. Given the hypercompact size of TnpB and IscB, they are excellent candidates for developing miniature gene editing tools that would facilitate in vivo delivery via AAV. To this end, our present study demonstrated the potential of TnpB for robust genome editing in both cultured cells and animal tissues. Although Kim et al. recently

reported engineering base editor from a 557-aa "TnpB"[23], both Siksnys and Doudna group lately demonstrated that "TnpB" used by Kim et al. study should be actually annotated as Cas12f1 that works as dimer unlike monomer TnpB[24,25]. Thus, our work was the first study to extensively show the rational optimization of TnpB to achieve excellent in vitro and in vivo performance for gene editing. Furthermore, we also showed the effectiveness of TnpB based gene editing therapy to correct fatal genetic disease in mouse model of tyrosineamia via in vivo single AAV delivery of TnpB and ωRNA. Interestingly, we performed stepwise truncation of cognate ωRNA to generate a ωRNA variant with short sequence and high efficiency. Our study represents a good start point to optimize TnpB or even IscB for more broad and convenient use in research and therapeutic scenario.

Endonuclease activity of TnpB was only shown with limited data in 2021 by Karvelis et al. study[1]. Extensive characterization of TnpB activity in mammalian cell and tissue were currently needed. Our finding corroborated the results from Karvelis et al. study, revealing unexpected higher activity of TnpB than that of Cas12f1 without further engineering. Moreover, we showed that deletion of 5'-end and partial internal sequence in ωRNA could enhance the gene editing performance of TnpB both in vitro and in vivo. Intriguingly, such deletion strategy was supported by two structural studies[26,27] of TnpB-ωRNA-DNA ternary complex published lately, suggesting the potential useful applicability of our ωRNA engineering strategy for more TnpB-like systems. In addition, the TnpB structure could accelerate the rational engineering of such compact enzyme with more demanding properties such as relaxed limitation of target-adjacent motif (TAM), enhanced editing activity and specificity etc.

Gene editing therapy was partly impeded by the limited AAV cargo capacity of only ~4.7 kb considering the fact that common Cas9, Cas12 and their derived base or prime editors have protein size beyond 1000 aa[3,28]. TnpB with less than 500 aa are highly desired gene editing enzymes for AAV delivery in vivo. Our results with TnpB in treating fatal tyrosineamia in mice signify the advantage of reducing gene editing cargo size despite the modest modification efficiency for *Hpd* target gene after TnpB-ωRNA optimization. Besides, compact TnpB size could permit using sophisticated regulatory sequences for switchable gene editing and reducing the AAV administration dose for high expression to enable safe therapeutic applications. Furthermore, our optimized ωRNA* variant with less than 100 nt would also be easy for synthesizing chemically modified ωRNA, which is very useful for ribonucleoprotein (RNP)-based gene editing applications. To investigate the anti-TnpB immunity in human population, we performed the ELISA and western blot analysis using human blood samples to find that there exists some but not all individuals without carrying antibody against TnpB, similar to SpCas9 (Supplementary Figs. 15 and 16). Therefore, to extend the applicability of TnpB for broad population in the future, it warranted the identification of more TnpB orthologs given the high diversity of TnpB in nature.

Overall, our study demonstrated the enhanced gene editing activity of TnpB via ωRNA engineering in cultured cells and showed its disease correction ability in animal models, indicating the potential of hypercompact TnpB-ωRNA system as effective miniature gene editing modality for more AAV-based disease treatment in animal models and even human patients.

## Methods

### Study approval

The objectives of the present study were to show proof-of-concept for in vivo TnpB-mediated gene editing in wild-type and disease mice. All animal experiments were performed and approved by the Animal Care and Use Committee of Shanghai Center for Brain Science and Brian-Inspired Technology, Shanghai, China. All animal experiments complied with the ARRIVE guidelines for reporting animal experiments. De-identified blood samples were obtained with the patients' written consent in strict observance of the legal and institutional ethical regulations and approved as a non-human study by the Institutional Review board of International Peace Maternity and Child Health Hospital, School of Medicine, Shanghai Jiao Tong University, Shanghai, China. No recruitment criteria was used and demographic information about our blood samples were also de-identified for the patient's privacy.

### Plasmid constructions

The pCBh-TnpB-hU6-BpiI plasmid encoded a human codon-optimized TnpB driven by CBh promoter, and hU6-driven ωRNAs with BpiI cloning site. The sgRNA and ωRNA were designed to be suitable for Un1Cas12f1, TnpB or other Cas proteins, then synthesized as DNA oligonucleotides and cloned into pCBh-Un1Cas12f1, pCBh-TnpB or other Cas plasmids to get the gene editing plasmids. Related plasmids were deposited to Addgene.

### Cell culture, transfection, and flow cytometry analysis

HEK293T were maintained in Dulbecco's modified eagle medium (DMEM) (Gibco, 11965092) supplemented with 10% fetal bovine serum at 37 °C and 5% $CO_2$ in a humidified incubator. For gene editing analysis, 1 μg TnpB or CRISPR plasmids and reporter plasmids were co-transfected using polyethylenimine (PEI) transfection reagent. After transfected cells were cultured for 48 h, we carefully resuspended the cell pellet, and then analyzed or sorted by BD FACSAria II for deep sequencing. Flow cytometry results were analyzed with FlowJo X (v.10.0.7).

### In vitro transcription of TnpB and ωRNA

TnpB mRNA was transcribed using the mMESSAGE mMACHINE T7 Ultra Kit (Invitrogen, AM1345). T7 promoter was added to ωRNA template by PCR amplification of pCX2280 using forward and reverse primers in the supplementary files. The PCR products purified with Omega gel extraction Kit (Omega, D2500-02) as templates were transcribed using the MEGAshortscript Kit (Invitrogen, AM1354). The TnpB mRNA and ωRNA were purified by MEGAclear Kit (Invitrogen, AM1908), eluted with RNase-free water and stored at −80 °C.

### Zygote injection and embryo transplantation

Eight-week-old B6D2F1 female mice were superovulated and mated with B6D2F1 male mice, and fertilized embryos were collected from the oviduct. The mixture of TnpB mRNA (50 ng/μL) and ωRNA (100 ng/μL) was injected into the cytoplasm of fertilized eggs using a FemtoJet microinjector (Eppendorf). The injected embryos were cultured in KOSM medium with amino acids at 37 °C under 5% $CO_2$ in a humidified incubator overnight and then transferred into oviducts of pseudo-pregnant 8-week-old ICR foster mothers at 0.5-d.p.c. (day post coitus).

### ωRNA engineering

At first, we performed the secondary structure prediction of ωRNA with online RNAfold webserver (http://rna.tbi.univie.ac.at/cgi-bin/RNAWebSuite/RNAfold.cgi). Based on the predicted structure, we divided ωRNA into six subdomains or segments and tested the influence of each segment on gene editing activity by deleting each segment one by one. In the first-round engineering, we found that deletion of segment #1, #2, and #3 (corresponding to ΔS1, ΔS2, and ΔS3 in Fig. 3b) does not affect gene editing efficiency of TnpB as shown in Fig. 2c, d. In the second-round engineering, we demonstrated segment #4, #5, and #6 (corresponding to ΔSL1, ΔSL2 and ΔSL3 in Fig. 2e) necessary for the normal gene editing activity of TnpB. In addition, substitution of terminal stem-loop structure with 5′-GAAA-3′ resulted in the final ωRNA* variant shown in Fig. 2g with improved gene editing efficiency and small RNA size.

### Targeted deep sequencing

To analyze TnpB gene editing efficiency, the DNA of successfully transfected cells or AAV-TnpB-ωRNA treated tissues were extracted with TIANamp Genomic DNA Kit (TIANGEN,) according to the manufacturer protocol. DNA was amplified with Phanta max super-fidelity DNA polymerase (Vazyme, P505-d1) for Sanger or deep sequencing methods. Deep-sequencing libraries were generated by adding Illumina flow cell binding sequences and specific barcodes on the 5′ and 3′ end of the primer sequence. The products were pooled and sequenced with 150 bp paired-end reads on an Illumina Hiseq instrument. FASTQ format data were demultiplexed using the Cutadapt (v.2.8)[41] according to the assigned barcode sequences. CRISPResso2 was used for gene editing analysis[29].

### PEM-seq analysis

Genome-wide off-target analysis was performed following PEM-seq protocol[17]. The 20 μg genomic DNA from TnpB edited or control samples were fragmented with Covaris sonicator to generate 300–700 bp DNA. DNA fragments was tagged with biotin at 5′-end by one-round PCR extension using a biotinylated primer, primer leftover removed by AMPure XP beads and purified by streptavidin beads. The single-stranded DNA on streptavidin beads is ligated with a bridge adapter containing 14-bp random molecular barcode, and PCR product was generated via nested PCR to enrich DNA fragment containing the bait DSB events and tagged with illumine adapter sequences. The prepared sequencing library was sequenced by Hi-seq 2500 with 150 bp pair-end reads. PEM-seq data analysis was performed using PEM-Q pipeline with default parameters.

### AAV virus production

The adeno-associated virus 8 (AAV8) serotype was used in this study. The TnpB plasmids with ωRNA was sequenced before packaging into AAV8 vehicle, and the AAV vectors were packaged by transfection of HEK293T cell with helper plasmids. The virus titer was $5 × 10^{13}$ (AAV-TnpB-ωRNA) genome copies/mL as determined by qPCR specific for the inverted terminal repeat.

### Gene editing treatment for tyrosinaemia mouse model

Mice were housed in a barrier facility with a 12-h light/dark cycle and 18–23 °C with 40–60% humidity. Diet and water were accessible at all times. The $Fah^{-/-}$ mouse model harbors the same homozygous G to A point mutation of the last nucleotide of exon 8, which result in exon skipping and the loss of FAH. $Fah^{-/-}$ mice were kept on 10 mg/L NTBC (Sigma-Aldrich, Cat. No. PHR1731) in drinking water when indicated. For viral particle injection, AAV8 ($4 × 10^{11}$ vg/mouse) in 200 μl saline were injected via the tail vein into 8–10 weeks old male and female mice. Mice were kept off NTBC water at 7 days post injection, and their body weights were recorded every 3–5 days. Mice were euthanized by $CO_2$ asphyxiation and harvested at 75 days after NTBC water withdrawal for histology and DNA analysis. Control mice off NTBC water were harvested when reaching >20% weight loss.

### Histological analysis, immunohistochemistry, and immunofluorescence

For histological analysis, liver sections embedded in paraffin were initially deparaffinized in xylene, followed by rehydration with a

gradient of ethanol ranging from 100% to 50%. Subsequently, the sections were washed in distilled water and stained with hematoxylin and eosin (H&E) and picrosirius red solution (0.1%) for histological examination.

Immunohistochemistry was conducted on deparaffinized sections, starting with the inhibition of endogenous peroxidase activity using a 0.6% hydrogen peroxide/methanol solution. Antigen retrieval was achieved using EDTA antigen retrieval solution. To minimize nonspecific binding, a 3% BSA solution was applied for 30 min, followed by overnight incubation at 4 °C with the primary antibody (HPD antibody, Santa Cruz, sc-390279, dilution 1:100). On the following day, slides were washed and incubated with the secondary antibody (Goat Mouse IgG, Abcam, ab97023, dilution 1:1000). Staining was visualized using the DAB peroxidase substrate kit.

For mouse tibialis anterior, diaphragm, and heart muscle tissues, immunofluorescence was carried out on frozen sections. Nonspecific binding was blocked with 5% normal goat serum in TBST for 1 h. The slides were then incubated overnight at 4 °C with primary antibodies (dystrophin antibody, Abcam, ab15277, dilution 1:100; laminin-2 antibody, Sigma-Aldrich, L0663, dilution 1:100) in 5% normal goat serum in TBST. Subsequently, slides were washed in TBST and incubated with secondary antibodies (Alexa Fluor 488 Goat Rabbit, Invitrogen, A-11008, dilution 1:1000; Alexa Fluor 594 Goat Rat IgG, Invitrogen, A-11007, dilution 1:1000) for 1 h. Imaging was performed using an FV3000 confocal microscope.

### Serum and urine biochemistry
Mouse blood was collected using retro-orbital puncture before mice were sacrificed. Mouse urine was collected every 24 h for a total of three times. Mouse plasma tyrosine levels were measured on a high-performance liquid chromatograph (HPLC1200) according to the standard protocols. ALT, AST, and bilirubin levels in plasma were determined using diagnostic ELISA Kits (Abcam). Succinylacetone levels were measured using High-Performance Liquid Chromatography (ACQUITY UPLC I-Class) coupled to tandem mass spectrometer (AB Sciex API 4000 LC-MS/MS).

### TnpB protein expression and purification
To express TnpB in *E. coli*, a prokaryotic TnpB expression plasmid (pCX2691) was constructed by cloning TnpB fragment in a pET2b-derived plasmid. TnpB expression plasmid was then transformed into Rosetta (DE3) competent cells for growth in LB medium with ampicillin (100 μg/ml) at 37 °C. After the OD600 reaches 0.6–0.8, TnpB expression was induced with IPTG. Induced cells were further cultured at 16 °C for 14–16 h. Next, we pelleted bacteria by centrifugation and resuspended in lysis buffer (20 mM Tris-HCl pH 7.0, 250 mM NaCl, 5 mM 2-mercaptoethanol, 25 mM imidazole, 2 mM PMSF and 5% (v/v) glycerol). After sonication, the cell lysate was centrifugated to remove cell debris, and the supernatant was load onto Ni2 + -charged HiTrap chelating HP column (GE Healthcare). TnpB protein was eluted with imidazole. Pooled TnpB fractions were dialyzed against storage buffer (20 mM Tris-HCl (pH 8.0 at 25 °C), 250 mM NaCl, 2 mM DTT and 50% (v/v) glycerol) and stored at −20 °C for further use.

### Human serum antibody analysis for TnpB and SpCas9 by ELISA
Adult serum was provided by Center for Reproductive Medicine, International Peace Maternity and Child Health Hospital, Innovative Research Team of High-level Local Universities in Shanghai, School of Medicine, Shanghai Jiao Tong University, Shanghai, China. About 2 ml of adult blood samples were collected from veins and left at room temperature for 30 min. Subsequently, they were centrifuged at 1800 × *g* for 10 min at room temperature, and the resulting supernatants were carefully transferred to tubes. The transferred samples were then subjected to a second centrifugation at 1300× *g* for 2 min and stored at −80 °C for future use.

The ELISA protocol was adapted from the standard methodologies[30,31]. In a concise summary, each antigen (TnpB, spCas9, human albumin) was coated onto a 96-well plate (0.5 μg/well) overnight at 4 °C in a coating buffer. Following this, the plates underwent five washes for 3 min each with TBST wash buffer. Subsequently, the plates were blocked with a 5% bovine serum albumin (BSA) blocking solution for 1 h at room temperature.

Serum samples were diluted 50-fold with 1% BSA Diluent Solution, added to the wells, and incubated for 1 h at 37 °C with shaking (200 rpm). Afterward, the plates were washed three times. Horseradish peroxidase-conjugated goat anti-human IgG Fc secondary antibody (Epigentek, cat#A-9000, dilution 1:5,000) was then applied and incubated for 1 h at room temperature. Following this, 3,3′,5,5′-Tetramethylbenzidine substrate solution was added and allowed to develop for 15 min before sulfuric acid was added to stop the reaction. The absorbance at 450 nm was measured using a SpectraMax i3X microplate reader.

### Immunoblot
For the detection of HPD expression in mouse liver, the murine liver tissues were incubated in RIPA Lysis and Extraction Buffer. Equal amounts of proteins were separated on SDS-polyacrylamide gel electrophoresis gels. Primary antibodies were listed as below: Anti-CRISPR-Cas9 (ET1703, HUABIO; dilution 1:1000), Anti-Human-IgG (ALPVHHs, dilution 1:1000), anti-HPD antibody (Santa Cruz, sc-390279; dilution 1:100 or 1:500), anti-P21 antibody (Abcam, ab109199; dilution 1:200), anti-Vinculin (Cell Signaling Technology, 13901 S; dilution 1:100), anti-dystrophin antibody (Abcam, ab15277; dilution 1:100), anti-laminin-2 antibody (Sigma-Aldrich, L0663; dilution 1:100). For anti-TnpB/SpCas9 antibody analysis, aliquots of 1 μg TnpB and 1 μg SpCas9 were resolved in buffer respectively and then applied to a 10% SDS-polyacrylamide gel electrophoresis. Samples were transferred to a PVDF membrane and blocked with 5% BSA in TBST for 1 h at room temperature. Immunoblots were then incubated overnight in TBST with 0.05% BSA with a 1:10 dilution of serum. Immunoblots were washed three times for 8 min in TBST on a shaker and then incubated with horseradish peroxidase-conjugated goat anti-human IgG Fc secondary antibody for 1 h at room temperature. After the incubation with Chemiluminescent substrates (#WP20005, Invitrogen), the membranes were viewed by the Image Lab™ Software 5.2.

### Statistical analysis
The number of independent biological replicates is shown in the figure legend. The data are presented as means ± SEM. Differences were assessed using unpaired two-tailed Student's *t* tests or one-way ANOVA. Differences in means were considered statistically significant at $P < 0.05$ (*$P < 0.05$. **$P < 0.01$. ***$P < 0.001$. ****$P < 0.0001$).

### Reporting summary
Further information on research design is available in the Nature Portfolio Reporting Summary linked to this article.

## Data availability
Deep-seq data are deposited to the GEO repository under accession number PRJNA973546. Source data are provided with this paper.

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

## Acknowledgements

We thank technical support from the laboratory animal center (Y.D., J.S., and T.Z.), optical imaging (L.T., K.S., and W.L.), and gene-editing core (R.Y., X.H., and X.Z.) facility in Shanghai Center for Brain Science and Brian-Inspired Technology as well as Lingang Laboratory. This work was funded by Lingang Laboratory (startup fund to C.X.), and Shanghai City Committee of Science and Technology Project (22QA1412300 to C.X., 20ZR1466600 to X.H.). S.M. was funded by National Natural Science Foundation of China (32100641). H.Y. was funded by National Science and Technology Innovation 2030 Major Program (2021ZD0200900) (H.Y.), Chinese National Science and Technology major project R&D Program of China (2018YFC2000101) (H.Y.), Strategic Priority Research Program of Chinese Academy of Science (XDB32060000) (H.Y.), National Natural Science Foundation of China (31871502, 31901047, 31925016, 91957122, and 82021001) (H.Y.), Basic Frontier Scientific Research Program of Chinese Academy of Sciences From 0 to 1 original innovation project (ZDBS-LY-SM001) (H.Y.), Shanghai Municipal Science and Technology Major Project (2018SHZDZX05) (H.Y.), Shanghai City Committee of Science and Technology Project (18411953700, 18JC1410100, 19XD1424400 and 19YF1455100) (H.Y.) and the International Partnership Program of Chinese Academy of Sciences (153D31KYSB20170059) (H.Y.).

## Author contributions

C.X., Z.L., and R.G. jointly conceived the project and designed experiments. Y.Z. and C.X. supervised the whole project. Z.L. and G.L. generated mouse model. Z.L. and R.G. designed vectors, performing in vitro experiments and scanning confocal imaging. X.H., X.S., X.C., and Z.S. assisted with the construction of plasmids and cell culture. R.Y. and X.Z. prepared AAV virus. R.G., Z.L., Z.S., Y.Y., X.L., and G.L. performed in vivo virus injection, tissue dissection, histological immunostaining, and liver function experiments. H.Z., X.C., J.S., S.M., and W.Z. performed PEM-seq experiments. M.Z. collected adult serum. Y.L. and Y.Z. performed bioinformatics analysis. R.G., G.L., and X.H. assisted with tissue dissection, immunostaining, and animal breeding. Z.L., R.G., G.L., C.H., Y.Z., and C.X. analyzed the data and organized figures. H.Y., C.H., Y.Z., and C.X. wrote the manuscript with data contributed by all authors who participated in the project.

## Competing interests

H.Y. is a founder of HuidaGene Therapeutics. The remaining authors declare no competing interests.
