## [Peer Review file · Nature Communications]

REVIEWER COMMENTS

Reviewer #1 (Remarks to the Author):

Due to its compact size and efficiency, the engineered TnpB- ω RNA system may provide a new and promising addition to the existing genome editing toolkit. However, I find several issues with experimental design and data presentation.

1. "First, we in vitro transcribed ω RNA that targets the mouse Tyr gene (Fig. 1b), and inject ω RNA together with TnpB mRNA into mouse embryos."

- Additional details on the injection and dosage are needed.
- n=27 for cas12f?
- Why not compare with saCas9?

2. Figure 1b.

- Axis typo.
- Please provide biological context on editing efficiency; i.e. how do you define it and what does it mean in the context of the Tyr gene and this experiment? e.g. if a single mouse has two copies of the tyr gene and only one is edited then that should be 50%; hard to grasp what a 40% edited mouse means

3. Further deep-sequencing for Tyr gene showed that 20% and 90% of indel mutations were induced by Un1Cas12f1 and TnpB, respectively (Fig. 1b).

- Is the data available? Fig 1b does not include any INDEL info.
- Could be good to comment on if there are multiple copies/alleles of Tyr and the extent of homo/heterozygous editing.

4. "Thus, the higher editing efficiency of TnpB as compared to Cas12f1 was largely due to its endonuclease activity."

- The endonuclease activity is not quantified and thus the above statement is not definitive.

5. TnpB exhibited an average of 90% editing efficiency for all six targeted loci in the Dmd gene

- How many animals were treated for each loci?

6. Most Figures.

- There are no */*** to show significance discussed in the caption...

7. Figure 2

- How does inducing a DSB here rescue GFP? DSB will introduce indels which doesn't guarantee proper frame of GFP returning? Further, introducing indels in the middle of a split GFP will mutate the sequence, leading to potentially nonfunctional gene?

8. In regards to ω RNA optimization, the authors provide little context on their methodology for segmentation of the ω RNA. How did the authors arrive at the subdomain replacements as described in Fig 2f.

9. Figure 2c: EGFP or GFP? RFP or mCherry? I have assumed this means %EGFP+ cells out of BFP+ and mCherry+ cells but maybe explicitly state this.

10. Figure 2a: How many days post transfection?

11. "To verify the reporter assay results for ω RNA*, we selected 14 endogenous genomic loci for further evaluation of gene editing performance in HEK293T (Fig. 3a)."

- What exactly are the loci? How did you select them? Could not find this information, even in the figures/supplement.

12. "Summary analysis of 14 loci also found significant".

- What is the "summary analysis"?

13. Figure 3c. The data shown and the p-value does not make sense. Can you provide the exact raw data and the corresponding statistical testing method to obtain the p-value?

14. Grammar changes, especially with past/present tense use: 143, 146, 157, 163, 192, 194

15. Line 206: What does NTBC and HPD stand for?

16. Figure S1 is missing RT-qPCR results.

17. Figure S4e: Based on the text (lines 182-185), it is unclear exactly what experiment was conducted. What is meant by "some loci that are barely edited"?

Reviewer #2 (Remarks to the Author):

Li and colleagues investigated TnpB for mammalian genome editing. TnpB is the ancestry endonuclease of CRISPR system. A clear advantage of TnpB is its small size (409 aa). They showed that the gene editing activity of native TnpB in mouse embryos is higher than previously identified small-sized Cas12f1. Then they engineered a noncoding RNA component of TnpB to increase the nuclease activity of TnpB. Finally, they showed that an optimized TnpB can be efficiently delivered to mouse livers in vivo by AAV8 to correct the biochemical phenotype of a mouse model of hereditary tyrosinemia type 1 (HT1). This is the first application of TnpB for embryo genome editing and in vivo genome editing. The biochemical correction of HT1 mice needs to be expanded with few additional measurements and controls.

1. Succinylacetone in urine and plasma has to be measured to confirm biochemical correction in HT1 mice.
2. Add serum tyrosine concentrations of wild-type mice and HT1 mice not treated with NTBC in Fig. S6.
3. Liver H&E in HT1 mice does not highlight fibrosis that is visualized by other stainings (e.g., Sirius red).
4. Western blot for HPD on liver extracts would be useful to support liver immunohistochemistry.
5. The results of the in vivo studies are convincing but a head-to-head comparison in HT1 mice with an AAV encoding the Cas12f1 would be important.
6. Are antibodies against TnpB less prevalent than antibodies against Cas? A lower prevalence of antibodies would be an additional desirable feature of TnpB over Cas.
7. The TnpB delivered by AAV is fused with EGFP. Therefore, the authors can easily check for duration of TnpB expression.

Minor issues

1. The authors repeatedly stated that the AAV injection in tyrosinemia type 1 mice prevented the disease phenotype, but the term 'prevention' is incorrect and it should be replaced by 'correction'.
2. Fig. 1b: there is a typo on the labeling of the y axis because they have 'fficiency' instead of 'efficiency'.
3. In the methods, the authors wrote 'for hydrodynamic liver injection'. Why hydrodynamic? Is this a typo?

Nicola Brunetti-Pierri

Reviewer #3 (Remarks to the Author):

In the manuscript Li, Guo, Sun, Li et al. provide proof-of-concept for in vivo TnpB-mediated gene editing in the wild type and disease mice. The work demonstrates the potential of TnpB nuclease as a genome editing tool. Authors first compared ISDra2 TnpB and UnCas12f editing efficiency by injecting mRNA directly into mouse embryo and showed that at the selected target TnpB outperformed UnCas12f. Next, they show that TnpB can be further improved by engineering of guide RNA. They also show that TnpB and guide RNA can be delivered into mouse liver using a single AAV vector.

Major concerns:

- P5-6. The section "Engineered TnpB-associated ω RNA with elevated editing efficiency" and the related figures require modification. In this section authors describe ω RNA modifications and show the predicted ω RNA structure (Fig. 2). Since experimentally solved structures have recently become available (Nakagawa et al., Nature, 2023; Sasnauskas et al., Nature, 2023), and these structures show significant differences compared to the predicted structure, authors should correlate the modifications with the experimental rather than predicted RNA structure to avoid confusion. The corresponding Figures have to be modified accordingly, replacing the predicted RNA structures with the experimental structures. My suggestion would be that authors show the predicted RNA structure that was used for the design of modifications and also provide the experimental RNA structure with mapped modification sites.
- The authors should clearly specify throughout the text that ISDra2 TnpB variant was used in the experiments since there are multiple TnpB variants characterized to date (Altae-Tran et al., Science, 2021; Karvelis et al., Nature, 2021; Nety et al., CRISPR J., 2023).
- It is not clear which RNA variant the authors used in PEM-seq and the liver disease model. If the RNA used was the wild-type (WT) variant, then the abstract is misleading because it suggests that only the "optimized" RNA was tested for AAV delivery. If the RNA used was indeed the wild-type variant, then the word "optimized" should be removed.
- How authors would explain that the editing efficiency of the extrachromosomal targets was lower than for the endogenous targets?

- HEK293 cell transformation conditions and reagents used should be provided in detail in the methods section both for the extrachromosomal and endogenous targets to enable other researchers to reproduce the experiments .

Minor comments:

- In the abstract TnpB size is indicated is 409, while in the figures it is 408.

- P2. In lines 52-54 (and P5, line 129), the authors state that "...247-nucleotides (nt) noncoding RNA...". However, it is unclear which variant they are referring to. Previous study has shown that the ω RNA length of ISDra2 TnpB is approximately 150-nt (Karvelis et al., Nature, 2021). Therefore, the authors should provide an explanation for their statement regarding the length of ω RNA.

- Materials and Methods. P24. In the section titled "Zygote injection and embryo transplantation," the authors should indicate the quantity of RNA that was injected.

- Materials and Methods. The authors should provide the sequences of the plasmids used in the study (benchling, etc.). This information is crucial for researchers in the field to be able to replicate the experiments.

- Figure S4. The dots colors in a-d panels should correspond to the colors designated for ω RNA variants in the e panel.

REVIEWER COMMENTS

Reviewer #1 (Remarks to the Author):

Due to its compact size and efficiency, the engineered TnpB- ω RNA system may provide a new and promising addition to the existing genome editing toolkit. However, I find several issues with experimental design and data presentation.

Response: Thank you for appreciating the significance of our study. The point-by-point response to all comments were presented as follow.

1. "First, we in vitro transcribed ω RNA that targets the mouse Tyr gene (Fig. 1b), and inject ω RNA together with TnpB mRNA into mouse embryos."

- Additional details on the injection and dosage are needed.

- n=27 for cas12f?

- Why not compare with saCas9?

Response: We injected the mixture of TnpB mRNA (50 ng/ μ L) and ω RNA (100 ng/ μ L) into zygotes collected from the oviduct of superovulated 8-week-old B6D2F1 female mated with male mice. A FemtoJet microinjector (Eppendorf) was used to inject TnpB- ω RNA mixture into the cytoplasm of zygotes. Detailed protocol was also added in the section of Material and Method in the revised manuscript as follow.

“Zygote injection and embryo transplantation

Eight-week-old B6D2F1 female mice were superovulated and mated with B6D2F1 male mice, and fertilized embryos were collected from oviduct. The mixture of TnpB mRNA (50 ng/ μ L) and ω RNA (100 ng/ μ L) was injected into the cytoplasm of fertilized eggs using a FemtoJet microinjector (Eppendorf). The injected embryos were cultured in KOSM medium with amino acids at 37°C under 5% CO₂ in a humidified incubator overnight and then transferred into oviducts of pseudo-pregnant ICR foster mothers at 0.5-d.p.c. (day post coitus).”

For Cas12f1-edited embryos, we obtained 13 offspring mice born after embryo transfer. The label of n=13 was added in the Figure 1b as follow (**Fig. R1**).

Fig. R1. Editing efficiency for Un1Cas12f1 and TnpB-edited mice by embryonic injection of gene editing RNA.

As suggested by the comment, we performed additional experiments to compare TnpB- ω RNA* with SaCas9 and SpCas9 for gene editing efficiency. Our results presented in the revised manuscript (Fig. 3c and Fig. S6) as follow (**Fig. R2**) showed that TnpB- ω RNA* exhibited comparable efficiency with SaCas9 and slightly lower activity than SpCas9.

Fig. R2. Gene editing activity comparison among TnpB- ω RNA, TnpB- ω RNA*, SaCas9 and SpCas9 in N2a (a,b) and HEK293T cells (c,d).

2. Figure 1b.
- Axis typo.

- Please provide biological context on editing efficiency; i.e. how do you define it and what does it mean in the context of the *Tyr* gene and this experiment? e.g. if a single mouse has two copies of the *tyr* gene and only one is edited then that should be 50%; hard to grasp what a 40% edited mouse means

Response: Thank you for the meticulous review. We have revised the axis typo in Fig. 1b as follow (**Fig. R1**). To help understand the editing efficiency presented here, we felt two important points to be clarified. First, the editing efficiency analysis was performed with the genomic DNA of tail tissue derived from the newborn mice generated by TnpB or Un1Cas12f1-injected zygotes, which could give rise to random percentage of edited and unedited cells in the mouse tissue depending on the efficiency of gene editing tools. Second, the gene editing tools injected in zygote stage could work from one-cell to multiple-cell stage in ex vivo embryos before in-uterus transfer depending on the half-lives of injected RNA as well as translated proteins, which would result in chimeric gene editing outcomes in mice born from injected embryos. If the gene editing tool could be highly efficient, it would produce homogenous editing or knockout of target gene with few chimerism in gene edited mice. As shown in Fig. R1, the low-efficient Un1Cas12f1 generate chimeric gene-edited mice with the majority of cells unedited, while the high-efficient TnpB generated homogenous gene-edited mice with *Tyr* knockout in early 100% cells. From deep-sequencing reads presented below (**Fig. R3**), we could observe the clear *Tyr* knockout results for Un1Cas12f1- and TnpB-edited mice.

Fig. R1. Editing efficiency for Un1Cas12f1- and TnpB-edited mice by embryonic injection of gene editing RNA

Fig. R3. Deep sequencing reads analysis for the representative Tyr gene-edited mouse by Un1Cas12f1 and TnpB

3. Further deep-sequencing for Tyr gene showed that 20% and 90% of indel mutations were induced by Un1Cas12f1 and TnpB, respectively (Fig. 1b).

- Is the data available? Fig 1b does not include any INDEL info.
- Could be good to comment on if there are multiple copies/alleles of Tyr and the extent of homo/heterozygous editing.

Response: We have added the INDEL results from representative mice edited by Un1Cas12f1 and TnpB in the revised Fig. S1 presented as follow (**Fig. R3**). For the deep-seq results in other figures, we also added the corresponding indel results in the supplementary figures (**Fig. S1, S2, S7, S8 and S9**). From the INDEL results presented in Fig. R3, both Un1Cas12f1 and TnpB injection generate multiple knockout alleles in Tyr gene but with one dominant allele for the individual mouse, which might be due to the persistent activity of injected gene editor in ex vivo embryos from one-cell to multiple-cell stage before in uterus transfer.

Fig. R3. Deep sequencing reads analysis for the representative *Tyr* gene-edited mouse by Un1Cas12f1 and TnpB

4. "Thus, the higher editing efficiency of TnpB as compared to Cas12f1 was largely due to its endonuclease activity."
 - The endonuclease activity is not quantified and thus the above statement is not definitive.

Response: We have revised the sentence as suggested.

"...Thus, the higher editing efficiency of TnpB as compared to Cas12f1 was largely due to its intrinsic activity....."

5. TnpB exhibited an average of 90% editing efficiency for all six targeted loci in the *Dmd* gene
 - How many animals were treated for each loci?

Response: In the revised manuscript, we have added the animal number in the Fig. 1d presented as follow (**Fig. R4**).

Fig. R4. TnpB-mediated gene editing efficiency for six exons in *Dmd* gene.

6. Most Figures.

- There are no */*** to show significance discussed in the caption...

Response: We felt sorry for the missing information on the statistical significance. In the revised manuscript, we have added the statistical information in both the caption and methods (*, $P < 0.05$. **, $P < 0.01$. ***, $P < 0.001$. ****, $P < 0.0001$).

“Statistical analysis

The number of independent biological replicates were shown in the figure legend. The data are presented as means \pm SEM. Differences were assessed using unpaired two-tailed Student’s t tests. Differences in means were considered statistically significant at $P < 0.05$ (*, $P < 0.05$. **, $P < 0.01$. ***, $P < 0.001$. ****, $P < 0.0001$).”

7. Figure 2

- How does inducing a DSB here rescue GFP? DSB will introduce indels which doesn’t guarantee proper frame of GFP returning? Further, introducing indels in the middle of a split GFP will mutate the sequence, leading to potentially nonfunctional gene?

Response: Two split GFP fragments were designed to carry homology sequence to facilitate homologous recombination (HR) repair after DNA double strand break (DSB) induction by TnpB between two split fragments. Therefore, the proper frame of our split GFP reporter could be generated by TnpB-induced DNA cleavage and cellular HR pathway. Our reporter design strategy was added in the revised manuscript as follow and the relevant paper (Yi Yang et al., 2016) was cited as well.

“...To facilitate screen of ω RNA variants, we designed a gene editing reporter with TnpB target DNA placed within a split and frameshifted GFP gene which could only be repaired via the single-strand annealing (SSA) pathway (Yi Yang et al., 2016) after disruption of TnpB target sequence to express GFP (Fig. 2a)....”

8. In regards to ω RNA optimization, the authors provide little context on their methodology for segmentation of the ω RNA. How did the authors arrive at the subdomain replacements as described in Fig 2f.

Response: At first, we performed the secondary structure prediction of ω RNA with online RNAfold webserver (<http://rna.tbi.univie.ac.at/cgi-bin/RNAWebSuite/RNAfold.cgi>). Based on the predicted structure, we divided ω RNA into six subdomains or segments and tested the influence of each segment on gene editing activity by deleting each segment one-by-one. In the first-round engineering, we found that deletion of segment #1, #2, and #3 (corresponding to $\Delta S1$, $\Delta S2$ and $\Delta S3$ in Fig. 3b) doesn’t affect gene editing efficiency of TnpB as shown in Fig. 2c and 2d. In the second-round engineering,

we demonstrated segment #4, #5, and #6 (corresponding to Δ SL1, Δ SL2 and Δ SL3 in Fig. 2e) necessary for the normal gene editing activity of TnpB. In addition, substitution of terminal stem-loop structure with 5'-GAAA-3' resulted in the final ω RNA* variant shown in Fig. 2g with improved gene editing efficiency and small RNA size. In the revised the manuscript, we have provided the detailed description in Methods section.

9. Figure 2c: EGFP or GFP? RFP or mCherry? I have assumed this means %EGFP+ cells out of BFP+ and mCherry+ cells but maybe explicitly state this.

Response: As suggested, we have changed the label to explicitly convey the percentage of EGFP+ cells out of BFP+ and mCherry+ cells.

10. Figure 2a: How many days post transfection?

Response: All the experimental analysis in Fig. 2 were conducted 48 hours post transfection, which was explicitly described in the revised Fig. 2a as follow (Fig. R).

Fig. R5. Reporter assay schematics of detecting cleavage activity of TnpB- ω RNA.

11. "To verify the reporter assay results for ω RNA*, we selected 14 endogenous genomic loci for further evaluation of gene editing performance in HEK293T (Fig. 3a)."

- What exactly are the loci? How did you select them? Could not find this information, even in the figures/supplement.

Response: These 14 endogenous loci were random selected from human genome to target intergenic regions. We have added the sequence information in the supplementary tables (Table S5).

"Table S5. Human 14 target sites sequences for TnpB activity evaluation in this study.

sgRNA	Target site sequences (5'-3')	oligos
Site1	TTTACACATCATCATATACA	TCAA TTTACACATCATCATATACA
		GGCC TGTATATGATGATGTGTA
Site2	AGAAGTGAGATGGCTCCAAA	TCAA AGAAGTGAGATGGCTCCAAA
		GGCC TTTGGAGCCATCTCACTTCT
Site3	GACCCAAAGAAATGTATTCC	TCAA GACCCAAAGAAATGTATTCC
		GGCC GGAATACATTTCTTTGGGTC

Site4	ATTCAAAAACACGCAAACCC	TCAA ATTCAAAAACACGCAAACCC GGCC GGGTTTGC GTGTTTTTGAAT
Site5	TCCTTTGCCAGGTTTCTGCA	TCAA TCCTTTGCCAGGTTTCTGCA GGCC TGCAGAAACCTGGCAAAGGA
Site6	TAATTAGAGCATAAATAAGA	TCAA TAATTAGAGCATAAATAAGA GGCC TCTTATTTATGCTCTAATTA
Site7	TCAAGTACCACCAGTTTTAT	TCAA TCAAGTACCACCAGTTTTAT GGCC ATAAAACCTGGTGGTACTTGA
Site8	AGAACACCCATAAGAACAAC	TCAA AGAACACCCATAAGAACAAC GGCC GTTGTTCTTATGGGTGTTCT
Site9	GAAAAGATTACAGAATCAGG	TCAA GAAAAGATTACAGAATCAGG GGCC CCTGATTCTGTAATCTTTTC
Site10	AGATGATGTTTCCACACATA	TCAA AGATGATGTTTCCACACATA GGCC TATGTGTGGAACATCATCT
Site11	CTGCCTTCAGAAAGCACCTT	TCAA CTGCCTTCAGAAAGCACCTT GGCC AAGGTGCTTTCTGAAGGCAG
Site12	AAAAATGCATGAAGCTCCTT	TCAA AAAAATGCATGAAGCTCCTT GGCC AAGGAGCTTCATGCATTTTT
Site13	TAAGGAACTAGAATCTAAAA	TCAA TAAGGAACTAGAATCTAAAA GGCC TTTTAGATTCTAGTTCCTTA
Site14	GAGTCCAGTCAGAAAGCAGA	TCAA GAGTCCAGTCAGAAAGCAGA GGCC TCTGCTTTCTGACTGGACTC

”

12. "Summary analysis of 14 loci also found significant".

- What is the "summary analysis"?

Response: We felt sorry for the confusing description. In the summary analysis, we aggregated the editing efficiency results for all 14 loci together to test the possible gene editing improvement of the ω RNA* variant compared to cognate ω RNA of TnpB.

13. Figure 3c. The data shown and the p-value does not make sense. Can you provide the exact raw data and the corresponding statistical testing method to obtain the p-value?

Response: The raw data was provided as suggested in the supplementary table (Table S1). The Fig. 3c was also removed as commented.

“Table S1. Gene editing efficiency for TnpB targeting human 14 sites.

Target site	TnpB- ω RNA			TnpB- ω RNA*		
Site1	80.9538	79.7051	77.9616	92.4128	92.7251	88.9477
Site2	91.1666	84.4706	86.6302	87.5109	91.692	87.0071
Site3	57.344	58.412	57.7154	83.7087	89.3263	78.8584
Site4	38.84	37.6139	42.7442	69.6037	72.8125	70.9723
Site5	12.5092	13.0002	19.1259	26.4571	31.0378	33.6545

Site6	54.4544	37.7141	50.0122	88.7723	90.357	88.4455
Site7	93.4096	96.0689	89.3796	91.7691	96.3689	95.9455
Site8	88.778	85.8337	86.537	91.9339	92.7217	88.5393
Site9	81.6767	91.7921	86.5068	86.3587	80.7439	81.0622
Site10	44.6191	52.511	51.8882	63.8667	64.4691	69.6262
Site11	40.9502	32.3023	36.2421	47.3455	42.452	49.8979
Site12	52.1795	51.2256	51.1079	41.4787	45.473	34.213
Site13	54.378	42.4125	53.2031	82.4258	81.97	78.2429
Site14	53.6541	61.4214	56.5062	83.8742	79.3327	81.1779

”

14. Grammar changes, especially with past/present tense use: 143, 146, 157, 163, 192, 194

Response: Thank you for helping improve the manuscript quality. We revised the corresponding sentences as suggested.

15. Line 206: What does NTBC and HPD stand for?

Response: NTBC and HPD stand for nitisinone and 4-hydroxyphenylpyruvate dioxygenase respectively, which are added in the revised manuscript.

16. Figure S1 is missing RT-qPCR results.

Response: The Fig. S1 in the previous version showed the sequencing results rather than qPCR for *Dmd* RNA after reverse transcription of RNA collected from muscle tissue of gene edited mice. We have revised the description in the caption to make it less confusing.

“Fig. S3. RNA genotyping of *Dmd*-edited mice with reverse transcription and targeted PCR by sequencing.

a-f. RNA genotyping results after reverse transcription and targeted PCR (RT-PCR) by sequencing for muscle from individual mouse edited by TnpB in exon 6, 20, 33, 43, 62 and 68 of *Dmd* gene.”

17. Figure S4e: Based on the text (lines 182-185), it is unclear exactly what experiment was conducted. What is meant by "some loci that are barely edited"?

Response: To avoid the confusion, we have revised the description as commented.

Reviewer #2 (Remarks to the Author):

Li and colleagues investigated TnpB for mammalian genome editing. TnpB is the ancestry endonuclease of CRISPR system. A clear advantage of TnpB is its small size (409 aa). They showed that the gene editing activity of native TnpB in mouse embryos is higher than previously identified small-sized Cas12f1. Then they engineered a noncoding RNA component of TnpB to

increase the nuclease activity of TnpB. Finally, they showed that an optimized TnpB can be efficiently delivered to mouse livers *in vivo* by AAV8 to correct the biochemical phenotype of a mouse model of hereditary tyrosinemia type 1 (HT1). This is the first application of TnpB for embryo genome editing and *in vivo* genome editing. The biochemical correction of HT1 mice needs to be expanded with few additional measurements and controls.

Response: Thank you for the positive comment on the significance of our study. We have responded to each comment in the following letter.

1. Succinylacetone in urine and plasma has to be measured to confirm biochemical correction in HT1 mice.

Response: Succinylacetone in urine and plasma were measured as suggested, which showed the biochemical correction of HT1 mice after TnpB- ω RNA treatment presented as follow (Fig. R6).

Fig. R6. Urine and plasma succinylacetone analysis for normal, untreated and treated HT1 mice.

2. Add serum tyrosine concentrations of wild-type mice and HT1 mice not treated with NTBC in Fig. S6.

Response: As suggested, we have added the serum tyrosine results for HT1 mice without both TnpB- ω RNA and NTBC treatment as control (Fig. R7).

Fig. R7. Serum tyrosine analysis for normal, untreated and treated HT1 mice.

3. Liver H&E in HT1 mice does not highlight fibrosis that is visualized by other stainings (e.g., Sirius red).

Response: We have performed Sirius Red staining to visualize fibrosis as commented, which is presented as follow (**Fig. R8**).

Fig. R8. H&E and Sirius red staining results for liver tissue of untreated and TnpB- ω RNA treated HT1 mice.

4. Western blot for HPD on liver extracts would be useful to support liver immunohistochemistry.

Response: We have performed western blot for HPD protein from untreated and TnpB- ω RNA treated mice as suggested to support liver immunohistochemistry results presented as follow (**Fig. R9**).

Fig. R9. Western blot for HPD protein from untreated and TnpB- ω RNA treated HT1 mice.

5. The results of the in vivo studies are convincing but a head-to-head comparison in HT1 mice with an AAV encoding the Cas12f1 would be important.

Response: In the revised manuscript, we added the head-to-head comparison results for HT1 mice treated with Cas12f1 or TnpB presented as follow (**Fig. R10**). It found that TnpB exhibited better therapeutic effect on preventing weight loss than Cas12f1 in the early stage after NTBC withdrawal, indicating the benefit of higher gene editing efficiency for the engineered TnpB- ω RNA.

Fig. R10. Comparison of therapeutic efficacy between TnpB and Un1Cas12f1 in HT1 mice. **a.** Body weight change for HT1 mice in different treatment groups. **b.** Body weight ratio for Un1Cas12f1 or TnpB-ωRNA versus TnpB-ωRNA* treated mice in 19-day after NTBC withdrawal.

6. Are antibodies against TnpB less prevalent than antibodies against Cas? A lower prevalence of antibodies would be an additional desirable feature of TnpB over Cas.

Response: As suggested, we performed antibody analysis for both TnpB and SpCas9 with several human blood samples. Our results presented as follow (**Fig. R11**) showed that there exists some but not all blood samples without carrying antibody against TnpB, as the scenario in SpCas9. To extend the applicability of TnpB for broad population in the future, it warranted the identification of more TnpB orthologs given the high diversity of TnpB in nature.

Fig. R11. Antibody analysis in the human blood for TnpB and SpCas9 using albumin as control. **a.** ELISA assay for anti-TnpB and -SpCas9 activity in human blood. **b.** Western blot for anti-TnpB and -SpCas9 activity in representative blood samples.

7. The TnpB delivered by AAV is fused with EGFP. Therefore, the authors can easily check for duration of TnpB expression.

Response: We have followed the TnpB expression until the sacrifice of treated

mice, showing the durable expression of TnpB in the entire treatment period presented as follow (**Fig. R12**).

Fig. R12. TnpB expression after in vivo delivery with AAV in three months.

Minor issues

1. The authors repeatedly stated that the AAV injection in tyrosinemia type 1 mice prevented the disease phenotype, but the term ‘prevention’ is incorrect and it should be replaced by ‘correction’.

Response: Thank you for the comment. We have changed the text as suggested to ‘correction’.

2. Fig. 1b: there is a typo on the labeling of the y axis because they have ‘fficiency’ instead of ‘efficiency’.

Response: This typo was corrected as commented as follow (**Fig. R1**).

Fig. R1. Editing efficiency of *Tyr* gene for Un1Cas12f1- and TnpB-edited mice by embryonic injection of gene editing RNA.

3. In the methods, the authors wrote ‘for hydrodynamic liver injection’. Why hydrodynamic? Is this a typo?

Response: Thank you for helping us avoid the typo in our manuscript. We have

revised it in the new version presented as follow.

Nicola Brunetti-Pierri

Reviewer #3 (Remarks to the Author):

In the manuscript Li, Guo, Sun, Li et al. provide proof-of-concept for in vivo TnpB-mediated gene editing in the wild type and disease mice. The work demonstrates the potential of TnpB nuclease as a genome editing tool. Authors first compared ISDra2 TnpB and UnCas12f editing efficiency by injecting mRNA directly into mouse embryo and showed that at the selected target TnpB outperformed UnCas12f. Next, they show that TnpB can be further improved by engineering of guide RNA. They also show that TnpB and guide RNA can be delivered into mouse liver using a single AAV vector.

Response: We appreciated the valuable comments by the reviewers. The point-by-point responses were carefully presented as follow.

Major concerns:

- P5-6. The section "Engineered TnpB-associated ω RNA with elevated editing efficiency" and the related figures require modification. In this section authors describe ω RNA modifications and show the predicted ω RNA structure (Fig. 2). Since experimentally solved structures have recently become available (Nakagawa et al., Nature, 2023; Sasnauskas et al., Nature, 2023), and these structures show significant differences compared to the predicted structure, authors should correlate the modifications with the experimental rather than predicted RNA structure to avoid confusion. The corresponding Figures have to be modified accordingly, replacing the predicted RNA structures with the experimental structures. My suggestion would be that authors show the predicted RNA structure that was used for the design of modifications and also provide the experimental RNA structure with mapped modification sites.

Response: Thank you for the constructive suggestion. We have revised the predicted RNA structure based on the experimental structural results published recently (Sasnauskas G et al., 2023; Nakagawa R et al., 2023), which was presented in the new version of Fig. 2 as follow (**Fig. R13**).

Fig. R13. Secondary RNA structure of different ω RNA variants in our study. **a.** Cognate ω RNA structure with 231 nt scaffold. **b.** Structure of engineered ω RNA-v1 variant with 117 nt scaffold. **c.** Structure of engineered ω RNA* variant with 99 nt scaffold.

- The authors should clearly specify throughout the text that ISDra2 TnpB variant was used in the experiments since there are multiple TnpB variants characterized to date (Altae-Tran et al., Science, 2021; Karvelis et al., Nature, 2021; Nety et al., CRISPR J., 2023).

Response: As suggested, we have explicitly in the revised manuscript specified the ISDra2 TnpB that was used in our study.

- It is not clear which RNA variant the authors used in PEM-seq and the liver disease model. If the RNA used was the wild-type (WT) variant, then the abstract is misleading because it suggests that only the "optimized" RNA was tested for AAV delivery. If the RNA used was indeed the wild-type variant, then the word "optimized" should be removed.

Response: PEM-seq was carried out with the engineered TnpB- ω RNA. For the liver disease model, we have performed additional experiment in the revised manuscript to compare gene therapy outcomes with both wild-type and optimized ω RNA variants, which were presented as follow (**Fig. R14**).

Fig. R14. Correction of fatal liver disease with in vivo delivery of TnpB-ωRNA via single AAV.

a. Diagram of AAV-TnpB-ωRNA/ωRNA* vector and gene therapy schematics in Fah^{-/-} mouse model of type I hereditary tyrosinaemia. **b.** Survival curve for disease mice treated with AAV-TnpB-ωRNA/ωRNA* or AAV-TnpB without ωRNA (TnpB only group). **c.** Body weight change during the observation period for disease mice in different treatment groups. **d.** Body weight ratio for TnpB-ωRNA or TnpB only versus TnpB-ωRNA* treated mice in 19-day after NTBC withdrawal. **e.** Histology analysis with H&E and Sirius red staining for mouse liver from different treatment groups. **f.** Western blot for HPD protein from untreated and TnpB-ωRNA treated HT1 mice. Data are represented as means ± SEM. A dot represents a biological replicate. Significant differences between conditions are indicated by asterisk. Unpaired two-tailed Student's t tests. * P < 0.05, *** P < 0.001, NS non-significant. Scale bars, 200 μm.

- How authors would explain that the editing efficiency of the extrachromosomal targets was lower than for the endogenous targets?

Response: Our reporter assay relied on both TnpB-induced DNA cleavage and cellular homologous recombination (HR) repair pathway, while gene editing of endogenous targets was induced by TnpB without depending on low efficient HR pathway in cells. Therefore, the editing efficiency of the extrachromosomal targets was lower than for the endogenous targets.

- HEK293 cell transformation conditions and reagents used should be provided in detail in the methods section both for the extrachromosomal and endogenous targets to enable other researches to reproduce the experiments .

Response: As suggested, we have provided detailed methods as follow for researchers to reproduce our results.

“Cell culture, transfection and flow cytometry analysis

HEK293T were maintained in Dulbecco's modified eagle medium (DMEM) (Gibco, 11965092) supplemented with 10% fetal bovine serum at 37 °C and 5% CO₂ in a humidified incubator. For gene editing analysis, 1µg TnpB or CRISPR plasmids and reporter plasmids were co-transfected using polyethylenimine (PEI) transfection reagent. After transfected cells were cultured for 48 hours, we carefully resuspended the cell pellet, and then analyzed or sorted by BD FACSAria II for deep sequencing. Flow cytometry results were analyzed with FlowJo X (v.10.0.7).”

Minor comments:

- In the abstract TnpB size is indicated is 409, while in the figures it is 408.

Response: We have revised the manuscript to make the statement consistent.

- P2. In lines 52-54 (and P5, line 129), the authors state that "...247-nucleotides (nt) noncoding RNA...". However, it is unclear which variant they are referring to. Previous study has shown that the ωRNA length of ISDra2 TnpB is approximately 150-nt (Karvelis et al., Nature, 2021). Therefore, the authors should provide an explanation for their statement regarding the length of ωRNA.

Response: In fact, Karvelis et al. provided the 231 nt ωRNA scaffold for gene editing in their previous study (Nature, 2021), which we used in our study accordingly for further engineering. With 20 nt spacer sequence, the wildtype ncRNA of ISDra2 TnpB will result in 251 nt full-length ωRNA. In the revised manuscript, we have explicitly described the ISDra2 TnpB and ωRNA length that are used in our study.

- Materials and Methods. P24. In the section titled "Zygote injection and embryo transplantation," the authors should indicate the quantity of RNA that was injected.

Response: We injected the mixture of TnpB mRNA (50 ng/µL) and ωRNA (100 ng/µL) into zygotes collected from the oviduct of superovulated 8-week-old B6D2F1 female mated with male mice. A FemtoJet microinjector (Eppendorf) was used to inject TnpB-ωRNA mixture into the cytoplasm of zygotes. Detailed protocol was also added in the section of Material and Method in the revised manuscript as follow.

“Zygote injection and embryo transplantation

Eight-week-old B6D2F1 female mice were superovulated and mated with

B6D2F1 male mice, and fertilized embryos were collected from oviduct. The mixture of TnpB mRNA (50 ng/μL) and ωRNA (100 ng/μL) was injected into the cytoplasm of fertilized eggs using a FemtoJet microinjector (Eppendorf). The injected embryos were cultured in KOSM medium with amino acids at 37°C under 5% CO₂ in a humidified incubator overnight and then transferred into oviducts of pseudo-pregnant ICR foster mothers at 0.5-d.p.c. (day post coitus).”

- Materials and Methods. The authors should provide the sequences of the plasmids used in the study (benchling, etc.). This information is crucial for researchers in the field to be able to replicate the experiments.

Response: As suggested, Plasmid sequences were provided as follow in the supplementary sequences as well as benchling links (<https://benchling.com/s/seq-0v72EZWTPKkMWi9XaljG?m=slm-w6LcjpDFmqrVsYzeJyIJ>).

“Supplementary sequences

>ISDra2TnpB

MIRNKAFVVRLYPNAAQTELINRTLGSARFVYNHFLARRIAAYKESGKGLTYG
QTSSSELTLLKQAEETSWLSEVDKFALQNSLKNLETAYKNFFRTVKQSGKKVG
FPRFRKKRTGESYRTQFTNNNIQIGEGRLKLPKLGWVKTKGQQDIQGKILNV
TVRRIHEGHYEASVLCVEVEIPYLPAAPKFAAGVDVGIKDFAIVTDGVRFKHEQ
NPKYYRSTLKRRLKAQQTLSSRRKKSARYGKAKTKLARIHKRIVNKRQDFLH
KLTTSLVREYEIIGTEHLKPDNMRKNRRLALSISDAGWGEFIRQLEYKAAWYG
RLVSKVSPYFPSSQLCHDCGFKNPEVKNLAVRTWTCPNCGETHDRDENAA
LNIRREALVAAGISDTLNAHGGYVRPASAGNGLRSENHATLVV

>ωRNA:

GATTCAAGAATCCCGAAGTGAAGAATCTTGCCGTCCGTACATGGACTTGC
CCGAAGTGTGGGGAAACCCATGACCGAGACGAGAACGCTGCGCTGAAC
ATTCGGCGTGAAGCGTTGGTGGCTGCGGGAATCTCAGACACCTTAAACG
CTCATGGAGGCTATGTCAGACCTGCTTCGGCGGGCAATGGTCTGCGAAG
TGAGAATCACGCGACTTTAGTCGTGTGAGGTTCAA

>ωRNA -v2(ωRNA*):

tGGTGGCTGCGGGAATCTCAGACACCTTAAACGCTCATGGAGGCTATgaaa
ATGGTCTGCGAAGTGAAGAATCACGCGACTTTAGTCGTGTGAGGTTCAA”

- Figure S4. The dots colors in a-d panels should correspond to the colors designated for ωRNA variants in the e panel.

Response: We have made revision as suggested for these results in the Fig. S10 presented as follow (**Fig. R15**).

Fig. R15. Characterization of gene editing activity for engineered TnpB-ωRNA system in mouse N2a cells.

a-d. Efficiency comparison using cognate and engineered ωRNA for mouse *Klkb1*, *Tyr*, *Hpd*, and *Pcsk9* gene editing. **e.** Summary statistic results for gene editing activity characterization of cognate and engineered ωRNA in N2a. Data are represented as means ± SEM. A dot represents a biological replicate. Significant differences between conditions are indicated by asterisk. Unpaired two-tailed Student's t tests. * P < 0.05, *** P < 0.001, NS non-significant.

REVIEWERS' COMMENTS

Reviewer #2 (Remarks to the Author):

Li and colleagues have addressed all the issues I previously raised.

Reviewer #3 (Remarks to the Author):

I went through the revised manuscript version. Authors addressed most of my concerns and I do not have any further comments.

Reviewer #4 (Remarks to the Author):

The previous concerns have been fully addressed. I agree to accept this manuscript for publication in Nat. Commun.